# Research and Development of Environmental Awareness about Water in Primary Education Students through Their Drawings

**Mª Paz Pozo-Muñoz** [1], **Carolina Martín-Gámez** [2,*], **Leticia Concepción Velasco-Martínez** [3] **and Juan Carlos Tójar-Hurtado** [3]

1   PhD student, University of Málaga, Málaga 29071, Spain
2   Department of Science Education, University of Málaga, Málaga 29071, Spain
3   Department of Research Methods and Diagnosis in Education, University of Málaga, Málaga 29071, Spain
*   Correspondence: cmartin@uma.es

**Abstract:** Drinking water is a scarce resource and the problems associated with its poor management and conservation are generating significant imbalances in the well-being of society, human health, and the environment. This research paper aims to determine the level of environmental awareness of primary school students in Spain (*n* = 95) of the management, use, and sustainable care of water, before and after applying a training program that allows working on the contents of water, from the different dimensions and shortcomings of environmental awareness identified in the paper. The study was undertaken from a mixed Research and Development approach (R&D), with an exploratory sequential design [QUAL(quan) → QUAL(quan)], in which, initially, a qualitative content analysis of the students' drawings was carried out, to subsequently apply a number of complementary quantitative analyses. The qualitative phase involved the elaboration of a mixed category (deductive-inductive) system that allowed for the organization and interpretation of the information obtained. The results of this work reveal low levels of knowledge about the integral water cycle. Primary school students also show a low degree of responsibility towards water issues. The conclusions of this study point to the need to address content related to water care and conservation in the educational curriculum from a multidimensional and interdisciplinary approach, in order to understand the origin and impact of water problems.

**Keywords:** water; problems; environmental awareness; primary education; program; drawings

## 1. Introduction

The COVID-19 pandemic has highlighted the fragility of our planet and the global challenges we face. One of them is the scarcity of water in the 21st century. The effects derived from climate change together with unsustainable consumption mean that this problem is intensifying more and more. Experts point out that this is generating a clear threat to the well-being of people, societies, and the environment. It is a problem that according to the researchers is the result of not having an environmental awareness about it. Therefore, the importance of changing the collective mentality and educating to create greater environmental awareness about water is highlighted. In other words, an environmental education is necessary, which does not have as its main objective the learning of concepts, but rather the search for the person's conscience to later go into detail in the forms of intervention [1]. For this reason, it is necessary to act urgently towards educating and training citizens so that they acquire, first, awareness about the water problem, and second, the skills with which to face them [2].

### 1.1. Socio-Scientific Problems as Teaching Contexts for Environmental Education

The present-day teaching of sciences involves great challenges in training students capable of developing in science and technology as competent 21st century citizens. This

idea is based on the need for all people to share responsibility for the tests faced by society in relation to the environment. Environmental problems (e.g., the greenhouse effect, water use and management, or energy dependence) are more pressing than ever, and there is a call for everyone to participate in the search for solutions. These are complex, global, and deeply interconnected challenges which require action mechanisms that must also be carried out both on a global and individual scale [3–5].

In this sense, since the beginning of the last century attention has been paid to the context of 'socio-scientific problems' (SSIs) and the opportunities they offer to promote the development of competencies related to the knowledge and skills that citizens should develop in their science and technology literacy process [6–9].

SSIs are considered to be those social problems in which science and/or technology are involved that are relevant to daily life and around which a large amount of controversy exists [10–12]. Science has not yet produced clear answers in many cases and, whatever position the individual or society adopts towards them, debate is not far away [13].

In addition, such problems are characterized by their interdisciplinarity [14] and, therefore, point to the integral formation of the person. Thus, for example, the problem of water and the knowledge associated with it involves different perspectives of study, including those of an economic, environmental, and sociocultural nature [15]. In other words, SSIs illustrate the complexity that occurs in situations where different agents of society are involved, and in the classroom, they can represent science learning contexts in which personal and social components play a relevant and motivating role [16,17].

Incorporating this approach to science teaching implies changing the traditional perception of science and adopting teaching strategies that open the door to those subjects involved in understanding the real problems of today's world [13]. In this sense, different authors argue that creating SSI-based learning contexts helps students develop values [18], and facilitates the learning of scientific content [19] among other aspects. It is about providing, from the teaching of science, training that contributes towards making students competent citizens for the near future so that, along with scientific and technological knowledge, there is an addressing of other elements that are also highly involved [20], and that promote, for example, values and attitudes such as social and environmental awareness [21].

In this context, ISSs are very appropriate educational contexts for the development of environmental education values and attitudes. Authors such as [22,23] argue that they provide opportunities for students to acquire a commitment to environmental issues and the necessary skills to protect and improve the environment by enabling them to examine and interpret it from a variety of perspectives (e.g., physical, biological, economic, ethical, and political). Therefore, the adoption of this methodology approach in environmental education is highly recommended, without forgetting to carry out an analysis and a reflection on what could be the best methodological strategies that would complete this approach and promote the improvement of environmental awareness.

*1.2. Teaching and Learning Strategies for Improving Environmental Awareness about Water*

Understanding the dynamics of water systems is increasingly important as it is linked to crucial environmental and social problems in our society such as climate change and drinking water scarcity [24], but how to do it to help students acquire environmental awareness related to these issues, and as [25] points out, so that they can understand the impact caused by man's action on the environment, and the influences and repercussions of their own actions on this?

In this sense, some authors such as [26] point out the importance of field trips or outdoor activities to improve the understanding of some key concepts to understand the water problem (e.g., phases of the water cycle). Other authors such as [27] posit the need to teach water-related topics in an accessible way using interdisciplinary and constructivist approaches, with innovative methods that emphasise hands-on training, interactive learning technologies, field trips, and activities linked to school life. Emphasising the constructivist approach, ref. [28]

recommends approaching water contents from the use of "transition hypotheses", where a progression from the simple to the complex is established in the organisation of contents and the reformulation of the problems investigated by students. This method makes it possible to detect the process of construction of the students' ideas and knowledge, to know their difficulties, and to guide the treatment of socio-environmental problems.

Along these same lines, ref. [29] also showed a good result related with the learning of another key concept for the environmental awareness of water (groundwater and aquifers) using a sequence of activities supported by constructivist theories and active learning. Specifically, this author used lectures with open-ended questions, worksheets, and interactive demonstrations. Regarding the latter, authors such as [30,31], furthermore showed the effectiveness of employing specialised software to support the learning of natural phenomena related to groundwater systems, and why and how the use of 3D immersive virtual reality give solid learning results on concepts related to the hydrological cycle [32].

On the other hand, and also related to the learning of concepts to develop environmental awareness about water, researchers such as [33] showed how the adoption of methodologies based on pictographic or graphic tasks, combined with semi-open-ended questions, are also highly appropriate strategies for teaching this issue, as well as highly appropriate diagnostic instruments for assessing the level of understanding of primary and secondary education students with regards to the elements and processes involved in the hydrological cycle. According to some authors [34–39], drawings are simple research tools that permit knowledge in relation to previous ideas, misconceptions, and mental models about scientific topics. In addition, these types of learning strategies become a great educational resource for finding out how students evolve in terms of their construction of scientific knowledge [40]. In fact, ref. [41] indicate that by comparing the evolution of student drawings and descriptions, it is possible to observe the degree of knowledge acquired, the modifications produced in their initial mental representations, and learning gaps. This was shown by studies such as [42] who found that through using a 'draw and tell' approach in combination with surveys, that the participants of their study seemed to have a superficial knowledge of water, little awareness of water saving, and limited knowledge of water resource conservation methods.

On the other hand, games are also suitable tools for simulating human interactions and decision-making in relation to environmental concerns such as water pollution. Proof of this are the results obtained by [43], who in his study highlights the contributions of game-based learning and presents the results of the application of a pedagogical tool entitled "The Hydrological Cycle Game", which allows students to understand different elements and processes of the water cycle; for example, the flow and transport of water or the cyclical and connected nature of the cycle, among others. In the same vein, ref. [44] show the positive effects of designing board and card games to educate and involve young people in understanding water problems, such as pollution, from a systemic perspective. These authors showed how, from the design of the board game "Pollutaplop", the students improved their water literacy, understanding the interactions between humans and natural systems while discussing, designing, testing, and modifying objectives, rules, and player actions. Ref. [45] also showed that the design of board games combined with simulation games for working on drinking water issues in a region of Bangladesh increased local stakeholder understanding of problems, solutions, and institutional actions on water management.

Another learning strategy relevant to encouraging the environmental awareness about water is the use of dilemmas. Thus, in their study, ref. [46] highlight the importance of value-based learning using 'ethical dilemma story pedagogy' (EDSP) as an educational resource for working on water issues. In this work, the dilemma of detergent waste being dumped directly into the river was raised, and the students, through this case study, understood the harmful effects that alkaline detergent residues have on the pH of the river and the resulting impact on the survival of species. These authors concluded that student participation in the analysis, evaluation, and resolution of ethical dilemmas facilitated the

understanding of the concepts and phenomena associated with water problems, expanding scientific knowledge on the subject, improving empathy, and putting into practice the principles of social justice [46]. Furthermore, along these lines, ref. [47] raises the possibility of working on water issues through "mapping controversies", with which it was intended that students addressed water problems as controversial curricular content from a socio-scientific perspective. This strategy proposed the analysis of news articles, involving the controversies or differences of opinions that appear among various actors or groups (citizens, companies, scientists, journalists, etc.). Through interpretation and reflection on the relationships map and the actions of these actors, students were able to better understand the phenomena and have a global understanding of water issues, to discuss on a socio-scientific basis, the value system, power relations, and socio-economic conflicts generated by the management and control of water resources [48].

However, despite all these methodological approaches for learning about water contents and encouraging the environmental awareness about this, some authors [49,50], point out that the textbook is still one of the most widespread educational resources in the teaching of this subject, even sometimes being a transmitter of conceptual errors, beliefs, or fragmented views on many of its contents (e.g., water cycle, irresponsible consumption, the economic value of water, or the problems of water resources pollution). In addition, given that environmental awareness is related to the children's perceptions and that the changes of these, in turn, are related to different factors such as age, education, geography, and socio-economical contexts [51], it is necessary to continue researching the development of programs, methodologies, and training resources to obtain data in different contexts to address the important socio-environmental issues related to water management and care.

*1.3. Research Aim*

This research paper aims to determine the level of environmental awareness of primary school students in Spain, before and after applying a training program focused on water management and care, taking the students' productions (drawings) as a reference. The specific objectives are:

-   To identify and analyse student conceptions and perceptions of water as an essential resource for living beings, from an ecocentric and anthropocentric perspective.
-   To know and value the knowledge students have about the phenomena, processes, and elements of the natural and urban water cycles.
-   To explore and interpret the respectful behaviours and attitudes shown by students towards water conservation and care.
-   To determine the interest and willingness of students to address water issues, assessing their knowledge and ability to identify the origin, impact, and possible solutions thereof.

## 2. Research Methodology

*2.1. General Background*

The study was conducted through a mixed Research and Development approach (R&D) [52], using a sequential exploratory design [QUAL(quan)→QUAL(quan)] [53,54].

These authors indicate that the design is based on a combination of a primary qualitative method (content of the drawings) and a complementary quantitative method (descriptive and comparative analysis). This design made it possible to show the greater presence or absence of the categories analyzed. The methodological approach included three phases of research. In the first phase, a content analysis of the 95 student drawings was carried out through the processes of reduction, arrangement, and transformation of the data, obtaining results and verification of conclusions [55].

I.   Data reduction. Through this procedure, the units of analysis (drawings) were segmented to be categorized and coded. The category construction process allowed the elaboration of a system of mixed categories, which included predefined categories (deductive), coming from the review of the specialized literature [56] and ad hoc

categories (inductive) that arose from the observation of sample drawing fragments. The mixed (deductive-inductive) category system was configured as follows:

- Macrocategory 1. Affective dimension.
- Macrocategory 2. Cognitive dimension.
- Macrocategory 3. Conative dimension.
- Macrocategory 4. Active dimension.

II. Data layout and transformation. Graphs, diagrams, and descriptive matrices were constructed to show the relationships between the categories and to discover the complexity of their structure.

III. Obtaining results and verifying conclusions. All the information was unified through processes of comparison, contrast, and triangulation to identify similarities and differences between the units of the categories. In the end, a comprehensive macro was built that contributed to the creation of a theory of environmental awareness. The Atlas.ti v7.0 program (2012) (Berlin, Germany) was used to support the qualitative analysis processes.

The category system meets the basic criteria of exclusivity, exhaustiveness, and unique classification principle. These types of assumptions "are framed in the line of authors who see in categorization a way of transforming textual data into data susceptible to measurement and quantitative treatment" [57–60]. Therefore, after the construction of the category system, they were quantified, and descriptive analyses (frequency and percentages) were applied to complement the interpretation of the qualitative findings found in the drawings.

In the second phase, a training programme was proposed as a development of the R&D to improve the level of awareness and training of primary school students in the management, care, and conservation of water resources. The programme covered the different dimensions and shortcomings of environmental awareness identified in the research. Table 1 shows the description of the activities that made up the proposal, together with the knowledge and dimensions of environmental awareness that were worked on in each of the sessions. The training programme continues to be applied in the same schools where the research was carried out and it is planned to extend its application to other educational centres in the province of Málaga (Spain).

**Table 1.** Description of training program activities.

| Session Description | Scientific Knowledge Covered | Environmental Awareness Dimension |
| --- | --- | --- |
| Session 1: Can we live without water? Students are presented with an imaginary situation in which they must explain to extraterrestrial beings what interest and relevance water has to our planet. The following questions are asked: can we live without water, what is water, what is it used for, where do we find it, in what processes is it involved, etc. The students are asked to draw a picture that answers these questions. | Previous knowledge | Affective, cognitive, conative, and active. |
| Session 2: Is there water for everyone? Students watch a documentary that highlights the problems of access, supply, and infrastructure of water resources in developing countries. It also analyses the current global dilemmas and challenges to ensure that water reaches all the inhabitants of the planet. The following questions are asked: could you live with only 3 litres of water per day; what would we need to know about to be able to solve the dilemmas raised in the documentary; what solutions can you think of to respond to the problems raised; and what can we do in our daily lives to improve the situation? Students should discuss as a group and present the conclusions reached to the rest of their classmates. | Social, economic, and environmental issues | Conative and active |

**Table 1.** *Cont.*

| Session Description | Scientific Knowledge Covered | Environmental Awareness Dimension |
|---|---|---|
| Session 3: Do we take care of the water around us? The following dilemma is posed to students: A coastal city, with strong periods of drought in summer and whose most important resource is tourism, sees its population tripled during the summer season. The mayor, in order to reduce costs, decides that water for human consumption is not to pass through the sewage treatment plant and is to be discharged directly into the sea. What do you think of this action? How do you think this affects the sea? What would you have done? What would be the solution? The students in small working groups should investigate and discuss, trying to answer the questions posed. The conclusions reached are shared with the class group. | Water collection, transport processes, storage, distribution, and water treatment processes. | Cognitive and active |
| Session 4: Do we take care of the water around us? The following dilemma is posed to students: Recently a law has been passed that limits the amount that can be fished daily on the coast of Málaga due to the disappearance of marine species that is occurring due to overexploitation. Because of the application of this law, a Málaga fishing family begins to suffer great economic difficulties to survive. The family decides not to abide by the law and fishes without respecting the established limits. What consequences do you think this situation has from an environmental point of view? Would you propose a solution? The students in small working groups should investigate and discuss, trying to give answers to the questions formulated. The conclusions reached are shared with the class group. | Social, economic, and environmental issues | Conative and active |
| Session 5: And the coast, do we take care of it? Students are told they are going on a trip to a beach near the school and asked to draw what they think they will find on this visit. The trip takes place, and the students collect photographs and notes in their notebooks of what they find on their coastal route. In the classroom, working in small groups, they contrast their initial drawings with their photographs and notes. Each group agrees on a drawing that gathers all the data collected during the outing. Finally, each group presents its drawing and proposes actions that could improve the ecosystem studied. | Terrestrial aquatic ecosystem and social, economic, and environmental issues | Affective and active |
| Session 6: Why is water important in my life? It is proposed that half of the students in the class take on the role of tourists and the other half take on the role of citizens of a coastal area. In this role, they reflect on why water is important and what actions they could take to look after it. After the group discussion, all groups present their consensus to the rest of the class. | Water Uses | Cognitive, affective, and active |

**Table 1.** *Cont.*

| Session Description | Scientific Knowledge Covered | Environmental Awareness Dimension |
|---|---|---|
| Session 7: Where does the water we use when we take a shower go? The students must give an answer to the question posed by means of a drawing. Next, a video describing the path of wastewater is shown. Students are required to carry out an experimental activity in which they use filtration and chlorination processes to treat several water samples with waste. Finally, the wastewater treatment process and its importance in the hydrological cycle is discussed. | Transportation processes, storage and distribution water and water treatment. | Cognitive |
| Session 8: What happens to the water we use when we take a shower? Students are taken to a wastewater treatment plant (WWTP) where some water pollution problems are discussed. At the end of the visit, in groups, students are required to create an advertising slogan to try to raise awareness of these problems and provoke a change in the behaviour of citizens. | Water treatment processes and associated pollution problems | Cognitive, conative, and active |
| Session 9: What is water used for? Students are divided into groups and assigned a plot of land next to a river in the province of Málaga. The students propose exploitation activities on their plots of land and study the water needs of each one. At the end, they explain their decisions in environmental, economic, and social terms. | Water uses and related problems | Affective and cognitive |
| Session 10: Can we live without water? As a final evaluation, students are again presented with the same imaginary situation as in session 1. Students are expected to modify and complete their initial drawings. | Final knowledge. | Affective, cognitive, conative, and active. |

In the third phase of this study an interpretation was made of the data obtained in the drawings made after the application of the programme (post-test). For this purpose, the same analysis processes were carried out as in the pre-test. Finally, a comparison of proportions (z-distribution) was made to observe possible significant differences between the drawings made before and after applying the programme.

*2.2. Participants*

For the selection of the participants in this study, purposive sampling was performed [61]. Ninety-five primary school students were selected, of whom 47 were in the 5th grade (49.5%) and 48 students were in the 6th grade (50.5%). The participants were aged between 10 and 12 years, of whom 55.8% were boys and 44.2% girls. Both groups came from public educational centre, located in a coastal neighbourhood of Málaga (Spain). The students' socioeconomic and cultural circumstances were low-medium.

The current and future state of the Málaga coast depends largely on how socio-environmental problems and sustainable practices are considered. In this sense, it is essential that young people understand the origin, the causes, and the impacts caused by the activity of human beings on water resources, so that they can contribute to providing innovative and appropriate solutions to their context.

*2.3. Information Collection and Analysis Instruments*

For the collection of information, an analysis and interpretation of the drawings made by the students was carried out. These drawings facilitated the assessment of student knowledge and environmental awareness of water management and care before and after applying the training program. Data collection was carried out through a guided activity: the students were required to reflect on a simulated situation in which they had to explain to extraterrestrial beings, through illustrations or drawings, the importance of water for life

on the planet. In order to encourage critical and creative thinking, the students were asked the following questions: What is water?; What is water used for?; Where do we find water?; What do you know about water pollution or scarcity?; and Do you think it is important to conserve and take care of water?

### 2.4. System of Categories

The following categorisation system constructed for the content analysis of the students' drawings is presented, taking into account macrocategories or dimensions of environmental awareness established by [56]. The macrocategories are described according to levels of knowledge acquisition (0, 1, 2 and 3) and learning phases (A, B, C, D, . . . ).

#### 2.4.1. Macrocategory 1. Affective Dimension

The affective dimension represents the students' perception of the importance of water for living beings. Within this macrocategory, ecocentric and anthropocentric awareness is assessed. Within ecocentric awareness, there is an analysis of whether water is a resource for the survival of humans only (Level 0-No Phases), humans and animals (Level 1-Phase A), humans and plants (Level 2-Phase B), and all living beings (Level 3-Phase AB). From the anthropocentric awareness, we assess whether water is perceived as a resource for direct use—food, hygiene, etc.—(Level 1-Phase A), indirect—industry, commerce, etc.—(Level 2-Phase B) or combining both direct and indirect uses (Level 3-Phase AB). Level 0 indicates an absence of the previous phases. Table 2 shows the relationship between the phases and the levels of the dimension.

**Table 2.** Relation between the phases and levels of the affective dimension.

| Level 0 | Level 1 | Level 2 | Level 3 |
|---|---|---|---|
| No phases are shown | Phase A | Phase B | Phase A-B |

#### 2.4.2. Macrocategory 2. Cognitive Dimension

The cognitive dimension assesses the students' knowledge of the natural and urban water cycle. This macrocategory assesses whether students know the properties of water and its surface distribution (Level 1-Phase A); the changes in the state of water and its subway distribution (Level 1-Phase B); and the origin, processes, and dynamics of the natural water cycle (Level 1-Phase C). Regarding the urban cycle, there is an analysis of whether they know the water collection processes (Level 1-Phase A); transport, storage, and distribution processes water (Level 1-Phase B); and the water treatment processes (Level 1-Phase C). Levels 2 and 3 refer to the acquisition of the learning phases in a combined way: level 2 (AB or AC or BC Phases) and level 3 (ABC Phases). Level 0 represents an absence of the previous phases. Table 3 shows the relationship between the phases and the levels of the dimension.

**Table 3.** Relationship between the phases and levels of the cognitive dimension.

| Level 0 | Level 1 | | | Level 2 | | | Level 3 |
|---|---|---|---|---|---|---|---|
| No phases are shown | Phase A | Phase B | Phase C | Phase A-B | Phase A-C | Phase B-C | Phase A-B-C |

#### 2.4.3. Macrocategory 3. Conative Dimension

The conative dimension addresses the level of pro-environmental responsibility that students have towards water management and protection. This macrocategory takes into account whether students are unable to identify that their habits or lifestyles affect water care (Level 1-Phase A) or, on the contrary, whether they have sufficient awareness to know that their daily actions have consequences on water quality and conservation (Level 2-Phase B). Level 0 is considered as an absence of the previous phases. Table 4 explains the relationship between the phases and the levels of the dimension.

**Table 4.** Relationship between the phases and levels of the conative dimension.

| Level 0 | Level 1 | Level 2 |
| --- | --- | --- |
| No phases are shown | Phase A | Phase B |

### 2.4.4. Macrocategory 4. Active Dimension

The active dimension represents the students' ability to identify the origin, causes, and possible solutions to water-related problems (scarcity, pollution, etc.). For this macrocategory, it is assessed whether the students have slight knowledge of the problems, showing a passive attitude (Level 1-Phase A). At a higher level, there is an analysis of whether they possess more or less extensive knowledge of the origin and/or causes of the problems from a socio-economic and social point of view, and also have a critical attitude (Level 2-Phase B) or whether they know the possible solutions to the problems and have a proactive attitude (Level 2-Phase C). If they present a combination of phases B and C, the students would be placed in level 3, where they not only know the origin and causes of the problems but also have a predisposition towards finding solutions to them (Level 3-Phase BC). Level 0 is valued as an absence of the previous phases. Table 5 shows the relationship between the phases and the levels of the dimension.

**Table 5.** Relationship between the phases and levels of the active dimension.

| Level 0 | Level 1 | Level 2 | | Level 3 |
| --- | --- | --- | --- | --- |
| No phases are shown | Phase A | Phase B | Phase C | Phase B-C |

### 2.5. Analysis of the Information

The information collected in the drawings was subjected to processes of data reduction, arrangement, and transformation in order to elaborate and verify the conclusions. To this end, techniques such as classification and categorisation were used to construct models and theoretical schemes to explain the reality studied. At the end of the process, explanatory matrices and graphs were developed [62] that allowed the creation of semantic networks that helped to represent the knowledge acquired by the students. To facilitate the qualitative analyses, the programme Atlas. ti v7.0 (2012) (Berlin, Germany) was used and for the quantitative analysis, the statistical analysis package SPSS v26 (Chicago, USA) was used.

## 3. Research Results

This section presents the results of the qualitative content analysis of the drawings and, subsequently, the descriptive quantitative analysis—frequencies and percentages—and a comparison of proportions (z-distribution) to show the most significant differences between the drawings before and after applying the programme.

### 3.1. Qualitative Analysis

The next section presents a qualitative analysis of the drawings based on the system of categories established to determine the level of environmental awareness of the use, management, and care of water among primary school students.

### 3.1.1. Affective Dimension

The following figures show two examples of the drawings made in the pre-test and post-test, corresponding to the ecocentric affective dimension of environmental awareness. Specifically, in Figure 1, it can be seen that the student draws the sea with fish; however, it is possible to verify its evolution given that in Figure 2 the same student is able to indicate that water is essential for the survival of more living beings and natural elements (mountains, clouds, trees, etc.).

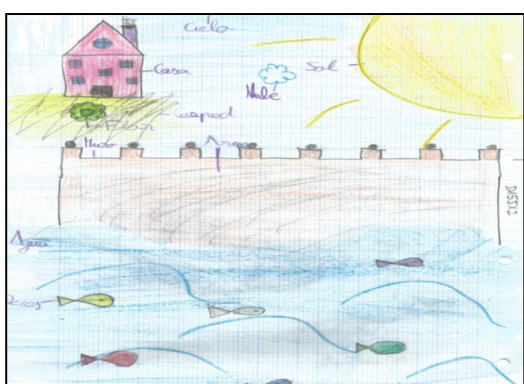

**Figure 1.** Example of Affective Dimension (pre-test). Ecocentric awareness. Level 1-Phase B.

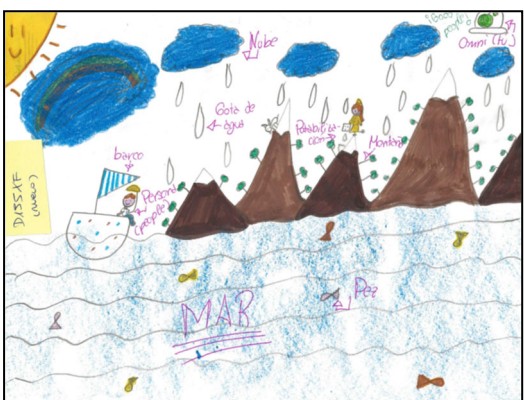

**Figure 2.** Example of Affective Dimension (post-test). Ecocentric Awareness. Level 2-Phase B.

Figures 3 and 4 show two drawings that illustrate the anthropocentric affective awareness of the students before and after applying the programme. As shown, before the programme the students were only aware of water as a resource for direct use—leisure, hygiene, irrigation, etc—and were at level 1 (Phase A) of environmental awareness (Figure 3). After the programme, the students draw some of the problems generated by industry due to poor management, and even propose possible solutions. The drawings illustrate a high level of environmental awareness on the part of the students in this dimension (level 2-Phase B). It should be pointed out that the study found no students who illustrated the direct and indirect uses of water together in their drawings.

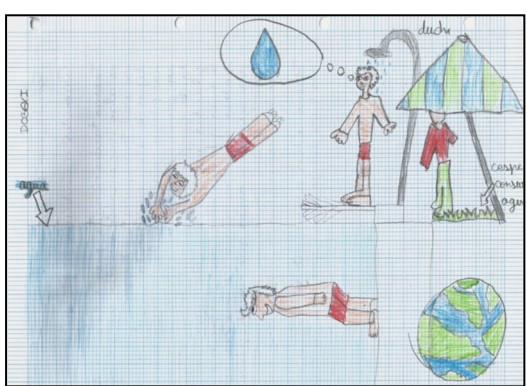

**Figure 3.** Example of Affective Dimension (pre-test). Anthropocentric Consciousness. Level 0-No phases.

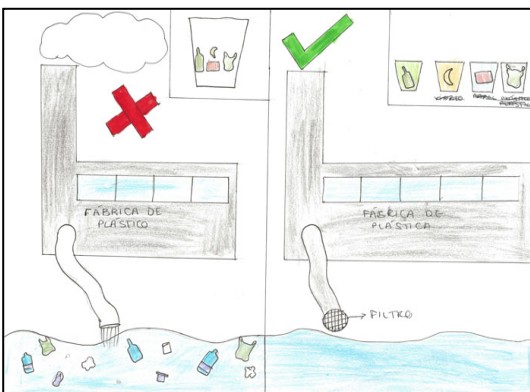

**Figure 4.** Example of Affective Dimension (post-test). Anthropocentric Consciousness. Level 2-Phase B.

### 3.1.2. Cognitive Dimension

The following figures illustrate the students' environmental awareness of the natural hydrological cycle before and after applying the programme. It is possible to appreciate how the students do not initially make any allusion to the phases of the natural cycle, being at level 0 (no phases) of environmental awareness in this dimension (Figure 5). However, after the programme, it is observed that the students are able to represent the changes in the state of water and its subway distribution, as well as the processes and dynamism of the natural cycle, placing them at level 2 (Phase BC) in the cognitive dimension of environmental awareness (Figure 6).

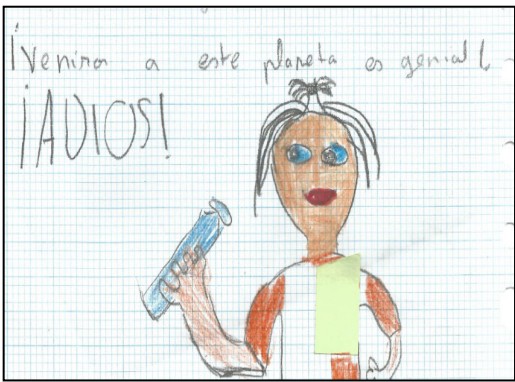

**Figure 5.** Example of Cognitive Dimension (pre-test). Natural water cycle. Level 0-No Phases.

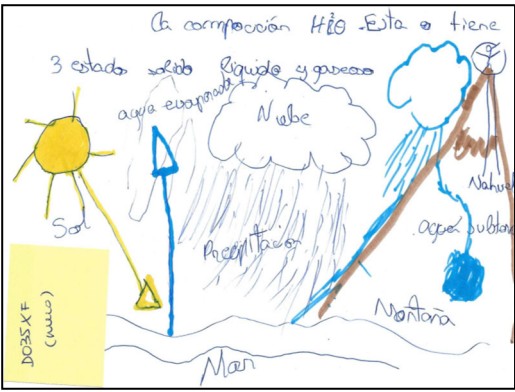

**Figure 6.** Example of Cognitive Dimension (post-test). Natural water cycle. Level 2-Phase BC.

Figures 7 and 8 show two examples of the cognitive dimension in reference to the urban cycle, before and after applying the programme. Specifically, Figure 9 shows a drawing that illustrates the students' knowledge of the natural water cycle, although it

does not mention any aspect of the urban cycle. As can be seen, the student is at level 0 (No phases), because he has only been able to represent some aspects of the natural cycle. However, it is possible to appreciate some evolution of the student since, as shown in Figure 8, he is able to draw the integral water cycle, including different processes and elements of the urban water cycle (although not all of them). In this sense, students would be at level 3 (Phase ABC), because they are able to illustrate the processes of catchment, transport, storage, and distribution, as well as the processes of wastewater treatment.

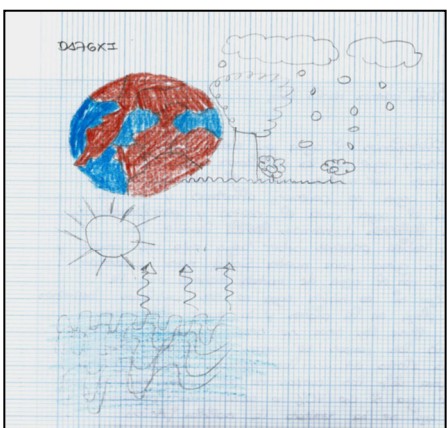

**Figure 7.** Example of Cognitive Dimension (pre-test). Urban water cycle. Level 0-No phases.

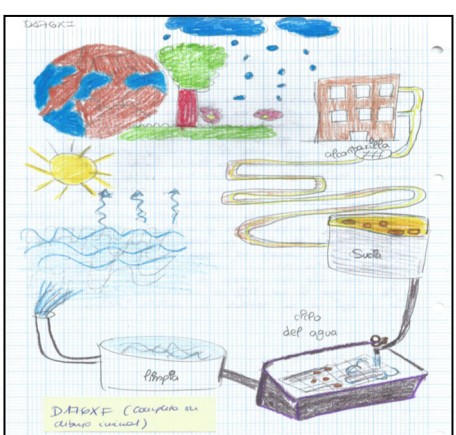

**Figure 8.** Example of Cognitive Dimension (post-test). Urban water cycle. Level 2-Phase BC.

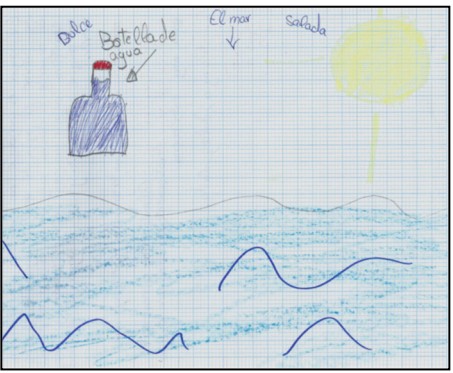

**Figure 9.** Example of Conative Dimension (pre-test). Level 0-No phases.

3.1.3. Conative Dimension

Figures 9 and 10 show two examples of the conative dimension of environmental awareness. In the former, it can be seen that the student does not represent anything related

to his habits or lifestyle and is therefore at a level 0 of environmental awareness in the conative dimension (Figure 9). On the contrary, Figure 10 shows that the student, after the application of the programme, seems to acquire a certain level of awareness about the impact of their harmful behaviour. For example, not recycling, bathing in the bathtub, or using fossil fuels are behaviours that the student shows in his drawing (Figure 10) and have an impact on the degradation of the environment (level 2, Phase B).

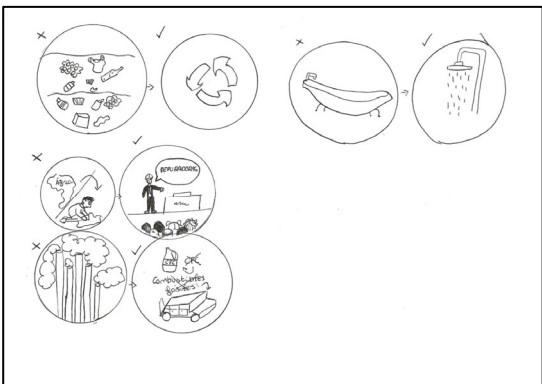

**Figure 10.** Example of Conative Dimension (post-test). Level 2-Phase B.

### 3.1.4. Active Dimension

Two examples illustrating the active dimension of environmental awareness are shown below. In Figure 11, it can be seen that the student does not present any awareness of water issues (scarcity, pollution, etc.) showing a level 0 (no phase) of environmental awareness in this dimension. On the contrary, after the application of the programme, it is possible to see that the students are aware of some problems related to water (Figure 12). For example, the drawing reflects the origin and causes of the problem of wipes that are improperly thrown into the sewage system and end up directly deposited in the sea. It also shows the problem of toxic spills such as oil from ships or radioactive substances from nuclear power plants, which equally end up in the sea. The drawings also show the proposal of some solutions by the student, such as using clean energies or improving the water treatment processes of sewage pumping stations. This student is therefore placed in level 3 (Phase BC) because he knows the origin, causes, and possible solutions to some of the problems related to water.

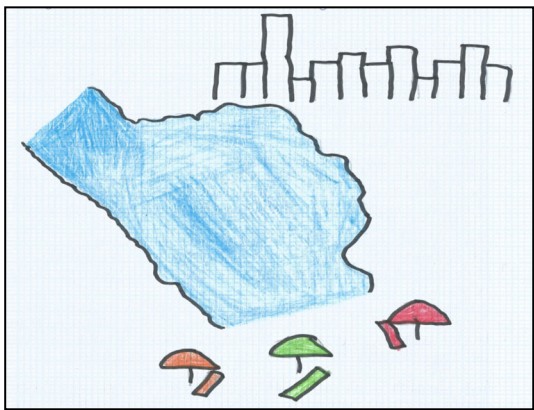

**Figure 11.** Example of Active Dimension (pre-test). Passive Consciousness. Level 0-No phases.

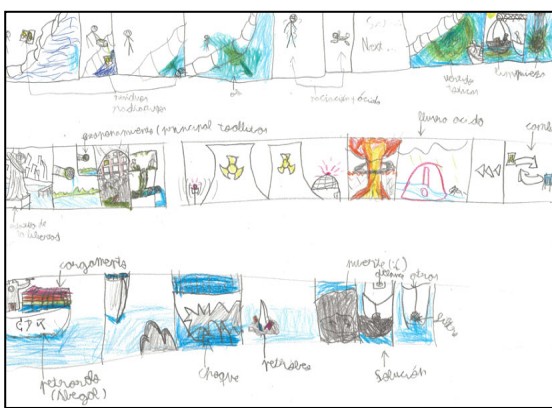

**Figure 12.** Example of Active Dimension (post-test). Critical Consciousness. Level 2-Phase B.

*3.2. Quantitative Analysis*

The results of the pre-test and post-test analysis of the dimensions of environmental awareness that were most significant for the study are shown below.

The following results show the references made by the students regarding the conative dimension, related to the responsible attitudes or behaviours that they show in their drawings towards water management and care. Figure 13 reveals that before the programme the students barely made references to the impact our lifestyle has on the conservation and care of water (No phases: 80.0% pre-test), and how this percentage decreases significantly after the application of the programme to 37.3% (z = 5.35, *p* < 0.0001). Significant improvements are also seen after the application of the programme with regard to the references made by the students about the importance of people taking responsibility for their actions (Phase B: 9.5% pre-test and 38.8% post-test, z = −3.68, *p* = 0.0001).

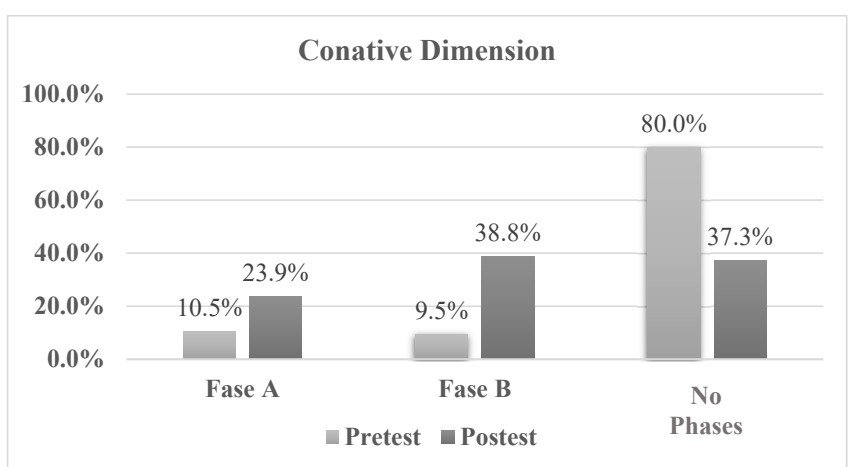

**Figure 13.** Results of the conative dimension.

On the other hand, the results of the active dimension (Figure 14) reveal the level of knowledge, interest, or predisposition of the students towards water issues (pollution, scarcity, etc.). This dimension shows a high percentage of students who do not make reference to water issues before the programme (No phases: 85.3% pre-test), and how this percentage decreases significantly after the application of the programme (No phases: 37.3% post-test, z = 6.01, *p* < 0.0001), improving the number of students who make reference to this dimension. On the contrary, there is a small percentage of participants who know the origin and impact of the problems and remain critical before the programme (Phase B: 4.2% pre-test), with the number increasing after the application of the programme (Phase B: 9.0% post-test). The fact that no students know the solutions or are proactive in the face of problems, either before or after the programme (Phase C: 0.0% pre-test and

post-test), is noteworthy. Nor do many students know the origin, impact, and solution to problems before the programme (Phase BC: 3.2% pre-test), although these results improve significantly after its application (Phase BC: 41.8% post-test, z = −4.84, *p* < 0.0001).

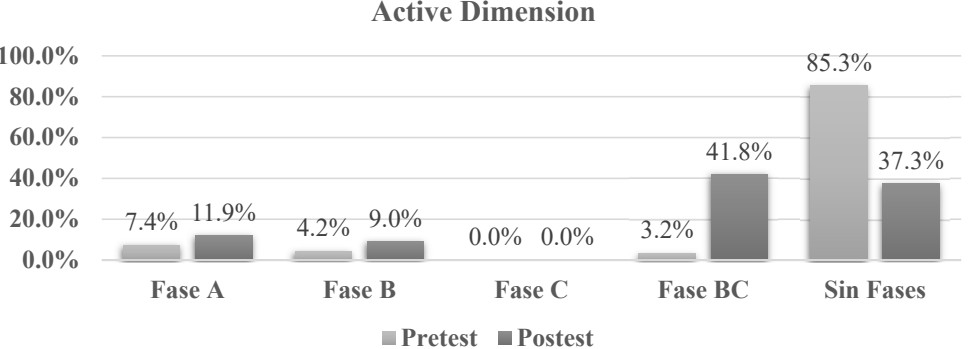

**Figure 14.** Results of the active dimension.

## 4. Conclusions and Discussion

This study shows the level of environmental awareness of students in the third cycle of primary education (10–12 years old) in Málaga (Spain), before and after applying a training programme on contents related to water management and care. In addition, the evolution of their knowledge of the phenomena, processes, and elements involved in the integral water cycle is studied. To this end, the students' drawings were analysed, considering four dimensions of environmental awareness (affective, cognitive, conative, and active), taking as a reference the theoretical model of [56]. This made it possible to discover and assess the level of knowledge, skills, and respectful attitudes shown by the students towards water conservation and care, valuing their interest and willingness to respond to water issues. In addition, it has allowed for the evaluation of the training programming applied, facilitating the identification of its strengths and weaknesses.

From the ecocentric perspective of the affective dimension of environmental awareness, the data, both before and after applying the programme, show that students reach a very low level of awareness of the importance of water for the life of all living beings. In the representation of the drawings, it can be seen how water only seems to be essential for human beings, without taking into account other living organisms. These results are consistent with those found in the study by [33], in which they found that less than 20% of the students included all the components of the biosphere (humans, animals, and plants) in their representations of the hydrological cycle. In addition, the aforementioned authors report that the students were also unable to include houses and factories in their drawings, a result similar to the one found in this study, where the students include very few elements related to indirect uses of water, linked to the products and services of economic activity (industrial, agricultural, etc.). These results are worrying if we consider that direct uses only represent between 1% and 4% of the total water footprint of a person, and the rest is due to indirect uses or virtual water [63]. In this sense, it is necessary to educate and raise awareness about the importance of changing our consumption model to reduce the direct and indirect water footprint and, as a consequence, minimise water scarcity problems that may affect our survival and the biodiversity of our planet [64]. Therefore, the training programme should aim to look at these issues in greater depth and delve more deeply into contents such as those raised by [65] who warn about the need to work with students on their conceptions regarding complex socio-scientific concepts such as water scarcity and virtual water-related problems.

In relation to the cognitive dimension, the results point to a certain lack of knowledge among the students about the phases of the integral water cycle. Regarding the natural hydrological cycle, the participants represented very little content related to changes of state, underground water movement, or the dynamic processes involved in the cycle both before and after applying the training programme. This may be due to the fact that, as [66] point

out, students do not present so many difficulties in recognising the processes involved in the cycle, although they do show problems in explaining them. Therefore, and as these authors conclude, the training programme should focus in greater depth on the water cycle in a contextualised manner, linking the process of water circulation with geographical locations and a lack of water, from a social, economic, and environmental perspective. In addition, within this dimension it is evident that students have problems in identifying where water can be collected and the hydraulic installations that allow its potabilization. They also have difficulties in understanding how water is distributed through the sanitation network and the purification processes carried out in wastewater pumping stations, which allow the resource to be returned to the sea in optimal conditions. Therefore, and in order to avoid, as recognised by [67], many students finishing their school education with significant gaps in basic knowledge of hydrology, it is necessary for educational programmes to emphasise the importance of clarifying the processes of the urban cycle [68] so that students do not confuse the water supply system that distributes drinking water with the wastewater system that disposes of contaminated water. It is therefore essential that these issues be addressed in greater depth in the training programme and planned outings to water treatment facilities.

Regarding the conative dimension that assessed the level of responsibility of the participants, the results show that the students do not initially seem to be aware that our lifestyle has an important impact on the conservation and care of water. Similar results were obtained by [67] in a group of 12-year-old students, who seemed to be unaware that water quality depends on its interaction with the environment and that pollution originates from multiple factors, including those of an anthropogenic nature. In this sense, after the application of the programme, the results obtained in our study seem to improve. In this way, and following the conclusions drawn by [67], it could be thought that having introduced awareness-raising activities on problems close to their environment in the training programme has helped to improve the level of environmental awareness of the participants in this dimension. Furthermore, using activities in which students can reflect on their own habits and what can be done to change them may also have been a strength of the training programme from the perspective of the level of responsibility. Authors such as [69] recognise that habits represent the greatest predictor for discovering the predisposition of people to conserve water, and that working on them can be of great interest for encouraging the acquisition of water care and conservation habits [70,71]. Furthermore, other authors [72–74] point out that there are other factors that influence the predisposition and interest of students to act responsibly, such as gender, identity, motivation, perception of self-efficacy, values, place of residence, socioeconomic level, educational stage, or educational policies in the formation of environmental awareness. Specifically, ref. [75] found in relation to gender, that girls seem to be more interested in environmental problems than boys, and even that girls are more participative when these issues are discussed within the family environment. Likewise, these authors also found significant differences between students from different regions (urban, rural, and mountainous). In this way, students from mountainous or sub-mountainous schools, compared with students who live in the capital or in agricultural areas, have a deeper knowledge of environmental issues directly related to the nature of their area (e.g., migration issues) and are more aware of the importance and ecological role of the area.

Regarding the active dimension that reveals the level of interest or predisposition that students have towards water issues, the results indicate that before the programme students made very few references in their drawings to the origin and consequences of several water problems such as pollution, water scarcity, or the overexploitation of aquifers. However, following the intervention, a notable increase can be seen in references regarding student knowledge of the origin, impact, and solution to the aforementioned water problems. These positive results may be due to the fact that the training programme provides an anthropogenic approach which, according to [76], is essential for students to understand the causes and processes of certain problems such as desertification, and thus to stop associating them only with natural factors. Moreover, another aspect to

highlight in the training programme that could be influencing this improvement in the active dimension of the participants' environmental awareness is the use of local examples on which to focus the dilemmas and activities proposed. In this sense, authors such as [77] indicate that teaching about water problems could be greatly improved by showing local examples (urban, school, neighbourhood) of inappropriate behaviours and how to solve them properly. This, as stated by these authors, has possibly allowed them to understand the impact that human actions have on our immediate environment. In addition, the fact that the programme has an interdisciplinary approach in which the subject matter is addressed from an economic, ethical, and social perspective, among others, could also be a strength. Along this line, ref. [77,78] highlight the value of interdisciplinary approaches for addressing important environmental issues. Despite this, the proposal of [79] is considered, which concludes that the repeated exposure to the negative impacts of human activities is insufficient for explaining environmental problems. It is therefore proposed that, for future implementations, the training programme also be required to include activities that not only encourage students to "learn to explain" but also to "learn to do". This will favour the development of real changes in their behaviour [77].

### 5. Limitations of the Study and Future Research

Although the purpose of this study was not to establish generalisations, it would be interesting to replicate the study by selecting a larger and more representative sample of students based on their socioeconomic level. In this sense, rural and urban high economic level primary schools could participate to research other contexts.

The findings of this study suggest that, from the methodological approach, the sample size may seem limited when considering it from the probabilistic point of view because it does not guarantee the representativeness of the data. In this sense, it is considered that an increase in the sample size also can improve the study of categorical relationships and obtain more conclusive results. However, from the qualitative perspective, the sample seems sufficiently representative to explain the study phenomenon and respond to the research objectives.

**Author Contributions:** Conceptualization, M.P.P.-M. and C.M.-G.; methodology, L.C.V.-M. and J.C.T.-H.; formal analysis, J.C.T.-H.; investigation, M.P.P.-M.; resources, M.P.P.-M.; data curation, L.C.V.-M. and J.C.T.-H.; writing—original draft preparation, C.M.-G.; writing—review and editing, C.M.-G. and L.C.V.-M.; supervision, C.M.-G. All authors have read and agreed to the published version of the manuscript.

**Funding:** This research received no external funding.

**Informed Consent Statement:** Informed consent was obtained from all subjects involved in the study.

**Acknowledgments:** We thank the study participants for their support, interest, and collaboration in the development of this research project. Moreover, we thank the R&D project, reference PID2019-105765GA-I00, entitled "Citizens with critical thinking: A challenge for teachers in science education", financed by the Spanish Government in the 2019 call.

**Conflicts of Interest:** The authors declare no conflict of interest.

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
