# Peer review of "Research and Development of Environmental Awareness about Water in Primary Education Students through Their Drawings"

_education, doi:10.3390/educsci13020119_

Round 1
Reviewer 1 Report
Congratulations for this paper which I think describe and analised a relevan topic "childreen´s evironmental perception". Just I only consider clarified some issues:
1) Environmental awareness is related to the children's perception and this perception changes depending on age, sex, educational context, geographical and socio-economical context, etc. I think is necessary to clarify this point in the theoretical background and discussion too.
2) In MPDI there are references related to your research which I recommend reviewing in order to complement the theoretical: Morón-Monge, H., Hamed, S., & Moron Monge, M. D. C. (2021). How Do Children Perceive the Biodiversity of Their nearby Environment: An Analysis of Drawings. Sustainability, 13(6), 3036.
Author Response
We have atacched a file that provides a point-by-point response to the reviewer’s comments.

Reviewer 2 Report
The central topic of the research article: to determine the level of environmental awareness of primary school students in Spain on the management, use and sustainable care before and after applying a training program that allows working on the contents of water, from the different dimensions and shortcomings of environmental awareness identified in the paper of water is of great interest although it has been approached from a very particular context and the results are not discussed with previous studies in this regard.
The introduction has few significant references. In some places it seems more like a manifesto or an opinion piece than an element of support for the arguments presented below. Once again, the topic is quite relevant, but it lacks more support and a better connection between the different sections of the document. In the introduction authors can explain their motivations and why the issue is relevant.
The literature review must be orderly: ideas are not connected. I suggest to group ideas and create a logic discourse.
The methodology of the research is weak, mainly because of the size of the sample (95 participants). The justification of the methodology is not robust (too generic). This election must be explained properly with research criteria.
Data collection and analysis must provide more insight in how the drawings were analyzed.
In general, the feeling is that there was no clear objective for this paper, since it does not clearly present a methodology for studying, neither the results of a study with s strong scientific support. My suggestions would be for the authors to describe carefully all the procedures undertook and deeply assess the conclusions and the validity of the study itself, with the small sample of people it includes as representative of a broader population.
Author Response

(The authors gave the same response as above.)
